# Towards an Integrated Methodology and Toolchain for Machine Learning-Based Intrusion Detection in Urban IoT Networks and Platforms

Denis Rangelov [1], Philipp Lämmel [1], Lisa Brunzel [2], Stephan Borgert [2], Paul Darius [1], Nikolay Tcholtchev [1,*] and Michell Boerger [1]

1   Fraunhofer Institute for Open Communication Systems (FOKUS), 10589 Berlin, Germany
2   Urban Institute (UI), Rössler Str. 88, D-64293 Darmstadt, Germany
*   Correspondence: nikolay.tcholtchev@fokus.fraunhofer.de; Tel.: +49-30-3463-7175

**Abstract:** The constant increase in volume and wide variety of available Internet of Things (IoT) devices leads to highly diverse software and hardware stacks, which opens new avenues for exploiting previously unknown vulnerabilities. The ensuing risks are amplified by the inherent IoT resource constraints both in terms of performance and energy expenditure. At the same time, IoT devices often generate or collect sensitive, real-time data used in critical application scenarios (e.g., health monitoring, transportation, smart energy, etc.). All these factors combined make IoT networks a primary target and potential victim of malicious actors. In this paper, we presented a brief overview of existing attacks and defense strategies and used this as motivation for proposing an integrated methodology for developing protection mechanisms for smart city IoT networks. The goal of this work was to lay out a theoretical plan and a corresponding pipeline of steps, i.e., a development and implementation process, for the design and application of cybersecurity solutions for urban IoT networks. The end goal of following the proposed process is the deployment and continuous improvement of appropriate IoT security measures in real-world urban IoT infrastructures. The application of the methodology was exemplified on an OMNET++-simulated scenario, which was developed in collaboration with industrial partners and a municipality.

**Keywords:** IoT; smart city; open urban platform; machine learning; cybersecurity; methodology; intrusion detection; toolchain

## 1. Introduction

In recent years, the number of devices connected to the Internet has increased dramatically. One field that is strongly affected by this trend is the Internet of Things (IoT) domain. The term IoT describes a group of physical, interconnected devices which interact with each other over a network without human intervention [1]. The rapid growth in the number of such devices, market penetration, revenue, and their integration with day-to-day life is expected to continue in the years to come [1]. More specifically, areas such as healthcare, smart grid, distributed energy sources (DER), self-driving vehicles (SDV), transportation, agriculture, smart environments, etc. will continue to experience radical transformation and improvements thanks to the opportunities provided by various IoT devices [1].

One of the major cornerstones for the successful integration and further growth of IoT is the implementation of proper security measures. While IoT devices bring various benefits to our day-to-day life, they can also imply major potential risks that can negatively impact the safety and well-being of the end users [2]. For instance, compromised healthcare infrastructure and hardware, self-driving vehicles, and home security cameras are illustrative examples for use cases where the lack of IoT security would lead to dire consequences and privacy violations [2]. By the same token, failing to provide security guarantees reduces the trust that users have in IoT networks. This can slow down and completely hinder

further IoT adoption and consequently could shut down the wide variety of benefits that the field introduces [1,2]. With this in mind, some of the most prevalent security challenges in the context of IoT are related to privacy, authentication and authorization, access control, and data storage and processing [1–3]. These areas are not specific to the IoT domain. However, there is a difference between traditional network security and IoT security. These differences stem from multiple factors such as the following [1]:

- There are noticeable software and computational resource limitations for IoT devices, which prevents the utilization of more sophisticated security algorithms;
- The IoT devices are low-powered, which restricts the usage of more energy-intensive security best practices and also increases the risk for technical failures (e.g., loss of data);
- The highly heterogenous hardware also leads to the usage of diverse software stacks and different data formats, which increases the available attack surface.

Challenges such as these introduce major security risks. Therefore, the goal of this work can be summarized as follows:

1. Analyze and identify the potential risks and available attacks against IoT- based platforms;
2. Present a comprehensive set of steps and measures that aim at providing improved security and attack prevention for a particular IoT-based platform and the underlying urban data platform—in this case, the UrbanPulse [4] of [ui!] [ui!] is the abbreviation for Urban Instutute GmbH—this is the industrial partner, with whom the case studies are being investigated and researched).

### 1.1. Open Urban Platforms

Open Urban Data Platforms are used to ease the use and analysis of urban data, which are mostly collected within an IoT network. The collected data are often combined with measurements and readings from other sources to produce valued services such as application or city management systems.

According to the German pre-standard DIN SPEC 91357 [5], an Open Urban Platform (OUP) is characterized as follows:

- They assist in the implementation of logical reference architecture following design principles of open APIs that supports data flows within and across city systems as well as enriching the raw data streams to generate smart data as required by the consuming entities;
- They exploit modern technologies to harvest, collect, and analyze urban data and provide the results to citizens and enterprises, e.g., sensor nodes and other IoT devices, cloud services, mobile connectivity, machine learning for analytics, and publishing and sharing via social media and APPs;
- They provide the building blocks that enable cities to rapidly shift from fragmented and isolated operation of individual infrastructures towards an integrated approach by connecting the systems via a platform, including cross-domain data analytics for predictions, forecasts, or better insight, and novel ways of engaging and serving city stakeholders offering smart services, both public and commercial.

### 1.2. Urban IoT Architectures and OUP

Open urban platforms often exist in certain integrated networks and architectures. These architectures usually differ within their structure. Therefore, an explanation of the used architecture and its construction—as depicted in Figure 1—is necessary:

- *Data sources and Actors:* IoT devices and sensors are utilized for collecting and analyzing data. The collected and analyzed data are transmitted to explicit gateways via, e.g., LoRaWAN or NB-IoT networks;
- *IoT Platform and Connectivity:* From the gateways, the data are forwarded across the network through different communication channels (e.g., mobile network cells) to IoT-

platforms. These platforms support the management of the IoT devices through their complete operational life cycle and are usually operated by the IoT device vendors;

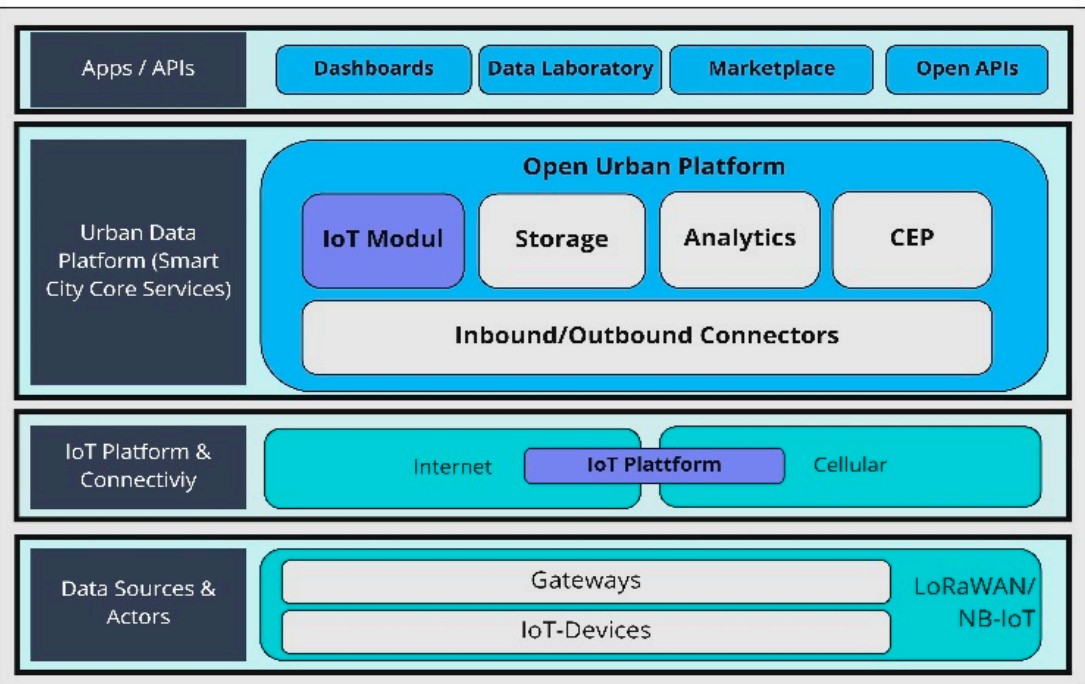

**Figure 1.** Abstract architecture of an open urban platform.

- *Urban Data Platform (UDP) (Smart City Core Services):* Connectors receive the data from the IoT-platform, normalize them, and usually enrich them by data stored in a UDP database. For example, a connector can receive a message with an IoT device ID and a date—in this case, the message on the output of the connector is extended with the geo-location of the IoT-device. In some cases, the IoT devices are connected directly to the UDP and are managed by an IoT module on the UDP. From the connectors the data are sent via a message bus to the storage and to a *Complex Event Processing* engine (CEP), which applies rules on the events and produces new messages. By doing this, the CEP can be considered as a virtual sensor and the new produced messages are stored in the *Storage* as well. The *Analytics* module combines analytic services and libraries. The analytic services are often machine learning/AI-based services, e.g., for predictions;
- *APP/APIs:* The UDP is connected to many APPs or provides outbound APIs. *Dashboards* or *Cockpits* are the most used APPs and provide information to users. Data laboratories use analytic services of the UDP and enable experts to perform sophisticated analysis. Marketplaces are used to provide data. Other data are provided by *Open APIs* on the outbound layer of the UDP/OUP.

### 1.3. Smart City Data Based Services

With the ongoing digitalization of cities, open urban platforms can be utilized in various fields [6]. The open provisioning of data can be seen as an opportunity to improve the urban living conditions and provide city services for different topics and areas of relevance:

- *Smart Government:* The key difference between an e-government and a smart government is the use of intelligently networked objects and cyber–physical systems [7]. Structures such as big data and open data are included in the development strategy. The change relates either to the resulting product, the process, or the prerequisites for the creation of the administrative service in question. Smart government therefore provides the means towards a data-driven digital administration [7];

- *Smart Economy:* Within a smart economy, cities are able to provide important data for new business models and to create conditions for economic development;
- *Smart Environment:* City data and their analytics provide new terms and conditions for environmental support. Within a large number of governmental tasks—such as waste monitoring or energy efficiency in the case of street lighting—data analytics can help to reduce $CO^2$ emissions and even provide an improved habitat for animals and humans;
- *Smart Urban Society:* Smart Urban Society addresses—in a data-based context—topics such as digital collective urban living and social interaction. Therefore, new incentives to live in a city can be developed and the social exchange can be supported. Topics such as smart health and education could also be the focus of this field and can be supported, e.g., through city data and self-sovereign entities;
- *Smart Mobility:* Smart mobility increases the use of environmentally friendly mobility options. Here, data are used to give users more information about the possibilities and benefits of using cheaper, faster, and environmentally friendly mobility solutions.

### 1.4. KIVEP Project

The KIVEP project is meant to research how the abovementioned concepts and technologies can be protected against cyber attacks in a smart city. The goal of the KIVEP project is to apply protocol anomaly detection techniques to the IoT base stations that connect the devices to the Internet via low-energy wireless networks such as LoRaWAN (Long-Range Wide-Area Network) or NB-IoT (NarrowBand IoT). The following problems are in focus: first, IoT base stations need insight into encrypted protocol packets for deeper protocol anomaly detection. However, breaking end-to-end encryption would give a successful attacker access to all traffic between the Internet and the connected IoT devices. This is why a privacy-preserving solution is being developed to monitor the data packets. On the other hand, only exemplary validation rules have been described for IoT protocols so far, which are not sufficient for effective protection against attacks from the Internet. For this reason, the automatic generation of comprehensive filtering rules for IoT protocols based on a new World Wide Web Consortium standard called "Web of Things Description" for describing IoT devices is being researched.

### 1.5. Contribution

The overall goal and clear contribution of the current paper is characterized by the following aspects:

- We proposed an integrated methodology for setting up and continuously improving cybersecurity solutions in urban IoT networks;
- We provided an overview of potential attacks on smart city IoT networks as a motivation for the abovementioned integrated methodology;
- We exemplified parts of the integrated methodology on an urban IoT network instance, which was simulated together with industrial partners and a municipality in Germany.

The novelty of our work is characterized by the devised methodology for planning and continuously improving the setup of cybersecurity solutions for urban IoT networks. Our approach was developed in close collaboration with industrial partners and a municipality, which intrinsically increases its relevance for real-world use cases. Furthermore, the approach was validated in an OMNET++ simulation in order to demonstrate its feasibility and capture feedback from relevant stakeholders (industry and utility companies) in the scope of the KIVEP project [8].

### 1.6. Structure of the Presentation

The rest of this paper is organized as follows. Section 2 presents the problem statement, for which the current work tried to find a solution. Thereby, Section 2 describes the general hazardous situation and elucidates the potential threats to urban IoT infrastructures. Section 3 continues with a more detailed discussion of the potential security issues

and summarizes available approaches to increase the level of IoT cybersecurity in urban ICT. The general issue that stands out is that all these single approaches do not represent an overall picture of how cybersecurity should be addressed in the smart city context for IoT networks. Hence, the following Section 4 defines and proposes such an overall methodology and process relating to how to initially set up cybersecurity solutions (with a focus on intrusion detection) in urban IoT and afterwards continuously improve and refine the installed mechanisms. Finally, Section 5 demonstrates key parts of the methodology based on a simulated IoT network use case, on which we worked together with a municipality and an industrial partner, whilst Section 6 draws conclusions and presents future research directions.

## 2. Problem Statement

This section focuses on briefly describing the general risks to which urban IoT networks are exposed. Furthermore, it prepares the reader for the possible realities of these risks in terms of potential attacks on urban IoT networks. These potential attacks are listed and described together with possible countermeasures in the following Section 3.

### 2.1. General Hazardous Situation

As mentioned previously, the number of IoT devices and the corresponding networks in which they participate grow at a rapid rate. The heterogenous nature and increased volume of devices lead to the development of new, previously unknown attacks and the uncovering of new attack surfaces. Due to their limited computational resources and energy capacities, IoT devices are not typically subjected to highly sophisticated security best practices, and they are often neglected as potential targets for malicious actors.

Nevertheless, the IoT networks still actively communicate with other traditional IP-based networks, which exposes them to common vulnerabilities and attacks. Furthermore, IoT sensors are used frequently to collect data in smart cities, healthcare, transportation, smart energy, and other domains where real-time decision-making is of crucial importance and can have severe consequences. Therefore, devising security mechanisms that preserve the privacy of end users while also protecting the critical network infrastructure must be a primary consideration in the context of urban IoT platforms.

### 2.2. Potential Attacks in Urban IoT Networks

As mentioned previously, IoT networks are a major target for potential attacks since they provide a wide attack surface and a highly diverse software stack. The potential dangers not only stem from commonly used attack vectors, but also from attacks specifically crafted and shaped against IoT network vulnerability points (e.g., energy capacity).

For instance, one common attack utilized in traditional IP-based networks is the denial of service (DoS) or distributed denial of service (DDoS) attack. In this case, the perpetrator floods the victim's system with a large volume of unwanted requests. This aims to cause damage in the form of inhibiting the system's ability to process legitimate requests or shutting the system down completely, which—depending on the system under attack—might lead to massive financial losses and in more extreme cases even to the loss of human life.

An example more specific to the IoT-domain is the so-called "node jamming" attack, which disrupts and/or completely prevents the transmission of signals generated by the IoT device. This can also be considered a form of a DoS attack, since it can shut down the functionality of a particular service and similarly to the DoS attack can have major negative implications. The examples given in this section are used as an abstract illustration for the potential dangers and attack vectors that can be exploited by malicious actors. However, a more comprehensive overview of IoT security's vulnerabilities and some of the available security countermeasures are presented in Section 3.

### 3. Discussion and Classification of IoT Attacks and Countermeasures

Similar to traditional networks, IoT architectures can be analyzed and evaluated according to a layer-based approach. There are multiple classification schemes proposed in the research literature, but most of them have the layers from Figure 2 in common. Based on these layers, we present different types of attacks and general defense strategies or architectural measures to protect urban IoT architectures. An overview of this classification is provided in Table 1. On the left we see the possible approaches/strategies for an attacker as abstracted from our literature review. These are also mapped to a corresponding layer in the IoT architecture in Figure 2 and combined with potential defense approaches and architectural measures.

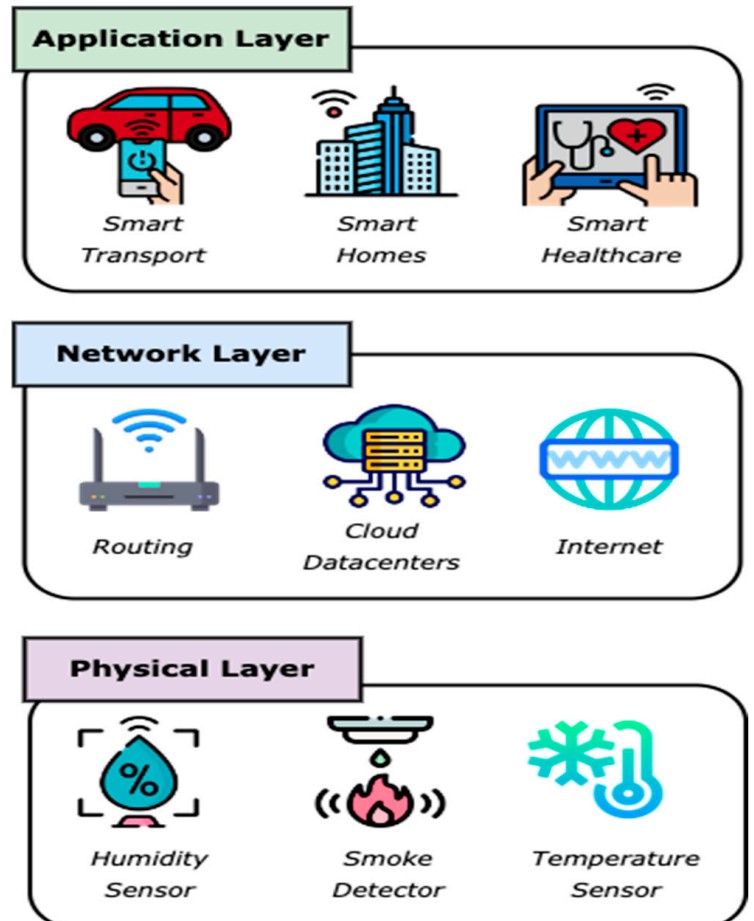

**Figure 2.** IoT layers classification (based on definitions from refs.) [1–3].

### 3.1. IoT Attacks

A detailed description of each IoT layer with the corresponding attack types and examples is provided in the following listing, which is followed by elucidations regarding the various architectural and algorithmic countermeasures—with concrete examples—that can potentially be applied to protect smart city IoT networks. The presented classification is based on a literature review, which aimed at finding common ground between the various classifications in the relatively new field of cybersecurity for smart city IoT devices.

**Table 1.** Classification of Attack Types and possible Defenses or Architectural Measures in Urban IoT Networks.

| Attack Type | Possible Defenses or Architectural Measures | IoT Layer Classification |
| --- | --- | --- |
| Data Theft | Blockchain, Edge and Fog Computing | Application Layer |
| Sniffing Attacks | Edge and Fog Computing | Application Layer |
| Malicious Code and Database Injections | Edge and Fog Computing | Application Layer |
| Distributed Denial of Service (DDoS) | Machine Learning and Deep Learning, Edge and Fog Computing | Network Layer |
| Spoofing Attacks | Machine Learning and Deep Learning, Edge and Fog Computing | Network Layer |
| Man-in-the-middle (MitM) | Machine Learning and Deep Learning, Edge and Fog Computing | Network Layer |
| Tampering | Edge and Fog Computing | Perception/Physical/Sensing Layer |
| Node Jamming or Radio Frequency Interference | Edge and Fog Computing | Perception/Physical/Sensing Layer |
| Sleep Deprivation or Denial of Sleep (DoSL) | Edge and Fog Computing | Perception/Physical/Sensing Layer |

**Application Layer:** The application layer serves as an interface between the end users and a given platform or service [1,2]. It provides functionalities such as authentication, authorization, data overview, and data access [2]. For this reason, the most common security vulnerabilities exploited at this layer are related to data theft and privacy violations [1]. For instance, some of the attacks performed at the application layer include the following:

- *Data theft* [1,2]: IoT devices are utilized in a wide range of use cases and, therefore, are involved in generating, processing, and transferring a variety of data. As pointed out by Hassija et al. [1], data that are being transferred are more vulnerable to attacks and consequently they can be stolen. At the same time, some of these data might include sensitive or private information. Therefore, if the end users cannot trust the IoT platform's privacy-preserving capabilities, they are unlikely to store their data on this platform [2]. Some of the common approaches for providing security guarantees against data theft comprise using data encryption, isolation, and network authentication [1];
- *Sniffing attacks* [2] occur when an attacker monitors the network traffic in an attempt to acquire sensitive user data [1]. The attack is executed by an attacker that uses malicious software to intercept and read confidential data flowing through the IoT network [2]. Similar to data theft, the prevention against such attacks includes the utilization of secure data transfer protocols [2];
- *Malicious code and database injections* describe attacks that are performed with the help of malicious user inputs such as scripts and code snippets. These attacks are possible due to insufficient code checks or the lack thereof [1]. The standard attack procedure includes an attacker finding a vulnerable entry point in the application layer and injecting a harmful piece of code that compromises the system [2]. Some of the common examples for such attacks include the SQL injection [9] and the cross-site scripting (XSS) attacks [1,10].

**Network Layer:** The main responsibility of the network layer is to handle the transmission of data coming from the physical layer across the IoT network [2]. Some examples of common network layer attacks include the following:

- *Distributed denial of service (DDoS)* is an attack that uses multiple devices or systems to flood a target service with unwanted traffic [1,11]. The main goal of the attack is to generate a massive number of requests which will either disrupt the normal functioning of the service or will completely shut it down. As pointed out by Liang and Kim [2], DDoS attacks are not specific to IoT networks. However, the large number of poorly secured IoT devices can become easy targets for a motivated perpetrator who can add the devices as a part of a botnet (e.g., Mirai [2,9,12]);

- *Spoofing attacks* take place when an adversary tries to fake their identity and impersonate a legitimate device or a user (e.g., by spoofing an IP address). This can give the adversary unauthorized access to certain resources or can allow them to observe and collect sensitive data transmitted over the network [1];
- *Man-in-the-middle (MitM)* is an attack during which an adversary is able to insert itself between two nodes in the IoT network. Consequently, the attacker is able to intercept, capture, modify, and relay data flowing between the two nodes without their knowledge [2]. More specifically, from the nodes' perspective it seems as if they are directly communicating with each other.

**Perception/Physical/Sensing Layer:** The perception layer is also known as the sensing [1] or physical layer since it is responsible for handling the physical IoT sensors and actuators. This layer is responsible for collecting data from the end devices and forwarding them to the network layer [2]. Some examples for devices that operate here include smoke detectors, camera sensors, and humidity sensors [1]. Exploiting these devices opens up opportunities for physical layer IoT attacks such as the following:

- *Tampering* refers to a physical intervention on the IoT device, through which the perpetrator modifies the hardware in a way that allows them to obtain sensitive information such as credentials, encryption keys, etc. [2];
- *Node Jamming* or *radio frequency interference* occurs when an attacker is near the location of the end devices and prevents them from successfully communicating with other devices on the IoT network [9]. This is achieved by sending noise signals that disrupt the wireless communication between the IoT devices [13];
- *Sleep deprivation or Denial of Sleep (DoSL)* is an attack during which the perpetrator targets low-powered IoT devices and tries to increase their power consumption in order to shut them down [1,14]. This is a form of DoS attack that can be achieved by injecting infinite (communication and computational) loops or modifying the hardware of the IoT device [14].

*3.2. Defenses and Countermeasures*

Given the importance and impact of the IoT domain on our day-to-day lives, it is crucial to explore the available defenses against the attack vectors described above. Similar to the attack classification, the available research literature classifies the defense mechanisms into one of the following categories.

**Edge and fog computing** are terms commonly used to describe two additional computational layers in the context of the cloud computing paradigm as described in Figure 3. Edge computing refers to computations taking place at the edge of the network, i.e., at the data source or very close to it instead of executing them in the cloud [11]. The main idea of edge computing is to reduce the data transfer between the cloud and the end devices. Instead, since the edge layer is in a very close proximity to and could even include IoT devices, there are faster data transfer times, low transmission costs, and near-real-time communication. This is essential for the implementation of well-established security best practices [1,15]. In addition, as pointed out by K. Sha et al. [15], the edge layer has more computational resources than the IoT end devices, which allows the utilization of more computationally expensive security mechanisms. These include not only encryption mechanisms such as homomorphic encryption, but also the implementation of firewalls, intrusion detection, and intrusion prevention systems at the edge layer, which can analyze and block incoming malicious traffic [15].

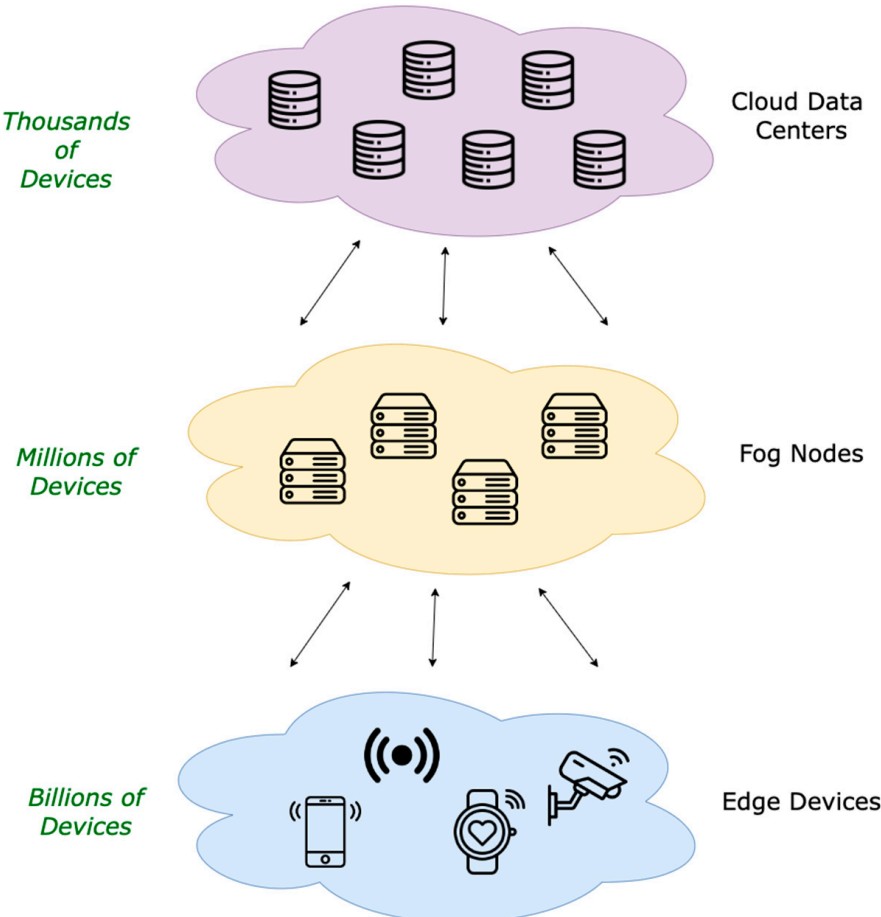

**Figure 3.** Cloud, fog, and edge computing hierarchy.

Furthermore, as discussed by Hassija et al. [1], since there is less data transfer and more local data processing, edge computing reduces the opportunities for data breaches and theft during transit. As mentioned above, the reduced data transfer also results in lower latencies and faster response times, which is not only important for the utilization of countermeasures, but it is also crucial for life-threatening scenarios such as the ones encountered in the health sector, automated vehicles, critical infrastructure, etc.

Within this context, the second important layer in the framework of the edge–cloud computing paradigm is fog computing. Fog computing describes computations taking place between the edge and the cloud layers (see Figure 3).

The main idea of fog computing is to serve as an additional layer between the edge and the cloud, which allows intermediate data aggregation, analysis, processing, and storage [1,16,17]. In this way, only the most essential data are further propagated to the cloud, which reduces the transmission costs, saves cloud storage space, and helps with performing real-time and time-sensitive tasks [1,16]. The devices most commonly used in the fog layer include switches, routers, and others [1,16].

In the context of IoT security, fog computing can address some common security challenges. For instance, since the fog layer typically aims at analyzing and filtering out malicious data, it can prevent anomalous traffic to be passed forward to the cloud or the main backend system [1]. Additionally, fog computing reduces the need for data transmission across the whole network, which decreases the risks for eavesdropping and data theft [1]. Finally, the fog provides an environment with higher computational resources compared with the edge devices. Therefore, fog nodes can implement more advanced security best practices and attacks targeting the resource-constrained IoT end devices are therefore harder to perform against the fog layer.

**Machine Learning and Deep Learning:** In recent years, the machine learning (ML) and deep learning (DL) domains have experienced significant growth and advancement and have become integral parts of a wide variety of industries. The IoT domain is no exception and ML/DL-based approaches can be applied for preventing and mitigating attacks, as well as for improving the security and privacy of IoT-based networks [1,10,18–21]. According to a comprehensive study presented by Al-Garadi et al. [10], some of the ML/DL-based methods most commonly used in the research literature can be classified as supervised, unsupervised, semi-supervised, and reinforcement learning methods (RL) [10]. These can be summarized as follows:

- *Supervised learning algorithms* are trained on data samples which are labeled and provide mapping between inputs and outputs. The most widely used supervised learning methods for IoT security include decision trees, support vector machines, Naive Bayes, K-nearest neighbors, random forest, deep neural networks (DNN), convolutional neural networks (CNN), recurrent neural networks (RNN), etc. [10];
- *Unsupervised learning* approaches try to identify patterns (typically by clustering) within an unlabeled data set. Common unsupervised learning methods used for improving the security in IoT networks include K-means clustering, principal component analysis, deep autoencoders (AEs), restricted Boltzmann machines (RBM), and deep belief networks (DBN) [10];
- *Semi-supervised learning methods* utilize a data set which typically contains a low volume of labeled and a large volume of unlabeled data points. Semi-supervised algorithms use both portions of the data for training, which places them in-between supervised and unsupervised learning [18]. The main advantage of these methods is that they can have improved accuracy due to the usage of a small number of labeled samples, while simultaneously being trained on a large volume of cheap, unlabeled data. Some of the semi-supervised methods used for IoT security include generative adversarial networks (GANs) and an ensemble of DNNs [10];
- *Reinforcement learning methods* train an agent which is supposed to make decisions based on the conditions present in a given environment. The agent is trained by interacting with the environment and receiving rewards proportional to the "accuracy" of its decision. Based on examples from research literature provided by Al-Garadi et al. [10], in the context of IoT security RL methods (e.g., Q-learning [1,10]), they are used primarily for preventing signal jamming attacks.

The methods described above can be utilized to prevent a wide spectrum of attacks performed at each layer of the IoT network. For instance, at the physical layer, user authentication strategies can be implemented with the help of DNN-based approaches that recognize users based on their daily habits [10,22]. As illustrated by the approaches summarized by Al-Garadi et al. [10], attacks on the network layer can also be detected with the help of ML/DL methods, e.g., malware and network anomaly detection can be performed with the help of autoencoders [23], whereas DoS and DDoS attacks can be detected with the help of deep neural networks [24]. Finally, at the application layer, various attacks (e.g., malware attacks [25], application layer DDoS attacks [26]) can be also successfully detected with the help of ML/DL approaches such as CNNs [25] and AEs [26].

**Blockchain:** The blockchain [27] is a decentralized ledger that stores data entries in a tamper-proof manner. It consists of blocks that are uniquely identified by so-called "hashes" and linked with each other with hash pointers. Therefore, modifying information inside the block (e.g., a transaction) changes its hash identifier, which invalidates all blocks in the chain that come after. In traditional blockchain implementation, new blocks are added to the chain by solving a resource-intensive cryptographic challenge called "proof-of-work". Since solving the challenge is computationally expensive and the blocks are secured with cryptographic hash functions, it is very unlikely that an adversary will be able to tamper with data stored in the blockchain. Additionally, each block stored in the chain is verified by all participants in the network and there is no central authority that can single-handedly

alter the transaction history or prevent transactions from executing. This set of properties makes the blockchain a compelling solution for some of the security challenges present in the IoT domain [19].

For instance, Dorri et al. [28] proposed a blockchain-based security solution for smart homes. The utilization of the blockchain for IoT security is challenging due to the low computational resources, high-latency transaction execution, and lower scalability [28,29]. Therefore, the authors introduced a blockchain-based solution that addresses these challenges. Additionally, the presented approach was evaluated with regards to multiple important security requirements: confidentiality, integrity, availability (also known as the CIA security triad), user control, and authorization. These requirements are fulfilled with the help of multiple techniques such as transaction logging into the blockchain, hashing, and symmetric encryption [28]. Furthermore, the proposed approach also serves as a defense against two common IoT attacks: DDoS and linking attacks (Linking attacks try to identify users within an anonymous environment by combining partial identifiers (e.g., zip code, gender, etc.) in an attempt to infer the complete user identity). In addition to this example, Hassija et al. [1] summarized some of the main benefits of blockchain security for IoT as follows:

- The blockchain can serve as a secure distributed data storage medium. The data stored in the blockchain are secured against tampering with the help of cryptographic hashing algorithms, and there is guaranteed data redundancy due to the absence of a single point of failure in the blockchain network.
- Nodes in the network are registered on the blockchain and therefore can be authenticated and identified, which prevents spoofing attacks.
- The blockchain serves as a decentralized alternative to traditional cloud servers. Centralized storage of information is a major target for perpetrators that want to steal sensitive data. Given that the cloud services provide shared infrastructure to many users at the same time, cloud storage can be compromised more easily compared with alternative blockchain-based approaches. Additionally, the data stored in the blockchain are distributed across all nodes in the network and signed (often also encrypted), which makes data theft attacks more difficult.

## 4. Methodology and Toolchain

The current work and the envisioned contributions are part of the BMBF-funded KIVEP project [8]. The project is carried out in a joined effort between the Urban Institute (ui!) and Fraunhofer Institute for Open Communication Systems (FOKUS). The main goals of the project include the research, analysis, and potential implementation of protocol anomaly detection in urban IoT networks.

In the following sections, we present the structure of a continuous process, see Figure 4, for achieving the objectives targeted within the scope of the KIVEP project. We also denote this methodology as KIVEP, which is the German abbreviation for "*Prevent and detect compromises of IoT devices through protocol anomaly detection*". (KIVEP stands for "Kompromittierungen von IoT-Geräten vorbeugen und erkennen durch Protokoll-Anomalie-Erkennung").

The KIVEP methodology is devised as close as possible to the needs and requirements of real-world deployments. This statement is based on the fact that we work in close collaboration with key players in the smart city IoT market in Germany as well as with the utility companies and municipalities, which are their direct clients in the context of different IoT deployments. As previously mentioned, this approach intrinsically increases the relevance of the below-presented KIVEP methodology for real-world use cases.

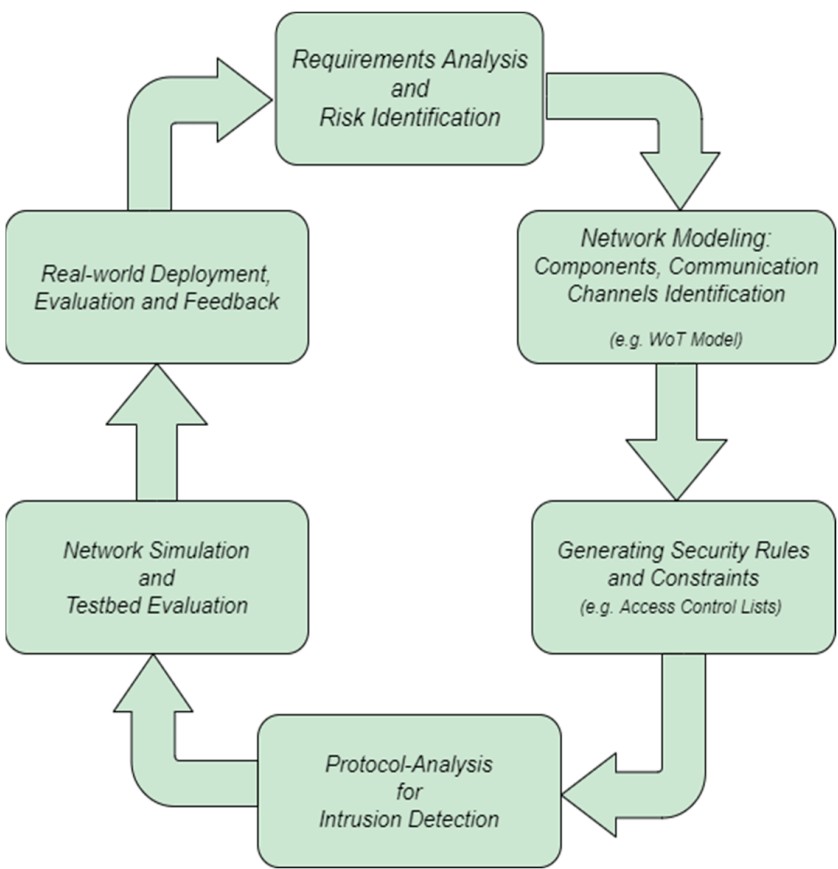

**Figure 4.** Structure of the KIVEP process pipeline and methodology.

*4.1. Requirements Analysis and Risk Identification*

This is the first step of the KIVEP design process described in Figure 4. Within this phase the various requirements (e.g., protocols, types of sensors, device parameters . . . ) for a particular urban IoT infrastructure are captured in the form of a requirement catalogue. Furthermore, the risks and potential threats for the system under design should be analyzed and captured in a corresponding risk and/or threat model using a tool-based approach. The captured requirements, risks, and threats should be systematically addressed during the design of the Smart City IoT infrastructure in question. Moreover, we expect that the architects, designers, and implementors will continuously gather experiences and lessons learnt throughout the different phases of the KIVEP methodology and will correspondingly update the requirements catalog as well as the risk and threat models.

*4.2. Network Modelling (e.g., WoT Modelling)*

Web-of-Things (WoT) encompasses a standardization scheme presented by the world wide web consortium (W3C). It provides guidelines that help dealing with the heterogeneity of the IoT domain by emphasizing the usage of well-established communication practices/structures/protocols, which can be utilized by a wide range of IoT and network devices.

For the purposes of the KIVEP methodology, we plan to utilize a WoT schema, which could potentially serve as a blueprint for the communication standards/structures utilized within a particular urban IoT network. The main idea is to create a model of the network that will help with the analysis of potential security vulnerabilities and possible attack vectors. The WoT model can be stored as a machine-interpretable file in JSON format that holds a description of the IoT Network and infrastructure under consideration. This model can be used for annotating and describing the places at which the security mechanisms (e.g., anomaly detection) assessed in the scope of KIVEP could be deployed. These are

theoretical considerations and the applicability of WoT will be further assessed throughout the course of the KIVEP project.

### 4.3. Security Rules and Constraints: Access Control List Generation

Access control lists (ACL) comprise a common approach in network and system management and represent a set of rules and constraints which determine the access permissions to a given system resource or an object. Resources commonly included in the ACL encompass routers, gateways, files, databases, etc. In the context of KIVEP, we plan to utilize an ACL for preventing unauthorized resource access and to filter out potentially malicious network traffic targeting the sensitive infrastructure of the urban IoT network in question. This can be achieved by identifying all resources and objects with the help of the analysis described above (see Sections 4.1 and 4.2). Then, as a next step, we intend to generate ACL rules for each endangered object or resource.

### 4.4. Protocol Analysis for Intrusion Detection

In this step of the pipeline, we planned the utilization of state-of-the-art ML and DL methods. Here, we focus primarily on analyzing the network packets and the corresponding traffic flow properties and not on the packet payload, i.e., DPI (Deep Packet Inspection) was not performed. The main focus of the protocol analysis conducted herein was placed on the application of unsupervised learning clustering methods (e.g., local outlier factor (LOF), autoencoders, etc.) for anomaly detection.

### 4.5. Network Simulation and Testbed Evaluation

The next step in the envisioned process pipeline is the design and development of a network simulation environment. As the name suggests, the main purpose of this simulation is to serve as a virtual representation of the network infrastructure and topology available in the production environment of a real-world urban IoT platform. In this way, the research efforts and the corresponding solutions proposed in the scope of this work can be evaluated with regards to their performance and their ability to provide the desired security guarantees. More specifically, the network simulation was developed as a testbed for the anomaly detection solutions examined throughout the KIVEP project, and it is structured as depicted in Figure 5.

The IoT devices in the diagram in Figure 5 serve as a representation of IoT parking lot sensors deployed on the premises of an industrial partner providing smart city solutions. The next important component is the so-called "middlebox" which is envisioned as a virtual environment that performs the function of intrusion detection (i.e., the protocol anomaly detection). The final location of the "middlebox" is not strictly established and might change depending on the needs of the project and the potential deployment challenges encountered along the way. After performing the protocol anomaly detection, the aim of the middlebox is to forward the analyzed traffic to back-end servers, where the data will be further processed and made available to the intended audience through the IoT platform and the sensor providers.

### 4.6. Real-World Deployment and Feedback

The final step planned within the continuous process of the KIVEP methodology is the deployment and evaluation of the proposed solutions in a real-life production environment. Assuming that the research efforts have resulted in the successful implementation of the desired anomaly detection method inside the OMNET++-simulated environment, the next and final step will focus on providing a real-world condition for testing the proposed solution. For these purposes, the anomaly detection methods implemented in the scope of KIVEP will be deployed in an urban IoT network, where their real-world performance will be evaluated.

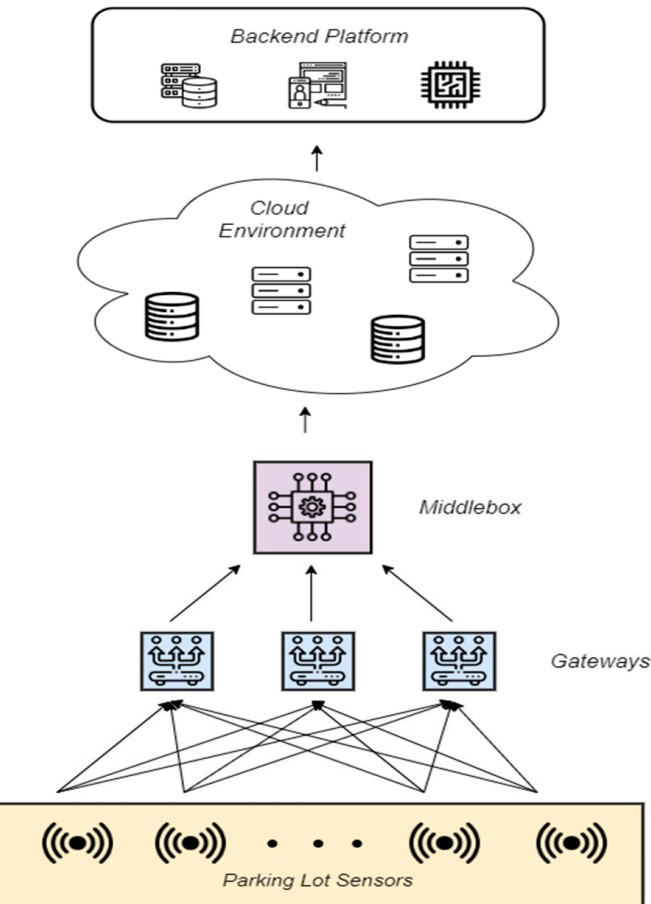

**Figure 5.** Architecture blueprint used by the network simulation.

The experiences from the real-world deployment and operations are submitted once again to the initial planning phase of requirements analysis and risk identification (see Figure 4 and Section 4.1) in order to enable the continuous improvement of the security solutions in place. Thereby, the process in Figure 4 is restarted, leading to a feedback loop to be executed during the operation of an urban IoT infrastructure.

## 5. Demonstrating the Methodology

Within this section, we showcase how the envisioned methodology could be applied on a real use case involving one of the associated partners in the KIVEP project. Given the sensitive nature of the data processed by the infrastructure provided by the related partner, we are not disclosing the exact setup utilized by them, but instead we transfer the patterns on a scenario which is mapped to the city of Berlin and to the surroundings of the Fraunhofer FOKUS institute.

### 5.1. Simulation Setup

Figure 6 shows the overall structure of the IoT network in question with its embedding in a smart city setting. The visualization is obtained after modeling the use case in OMNET++ [30] and shows 16 sensors as the KIVEP partner deploys them in the real case. However, the map behind shows a geolocation of Berlin.

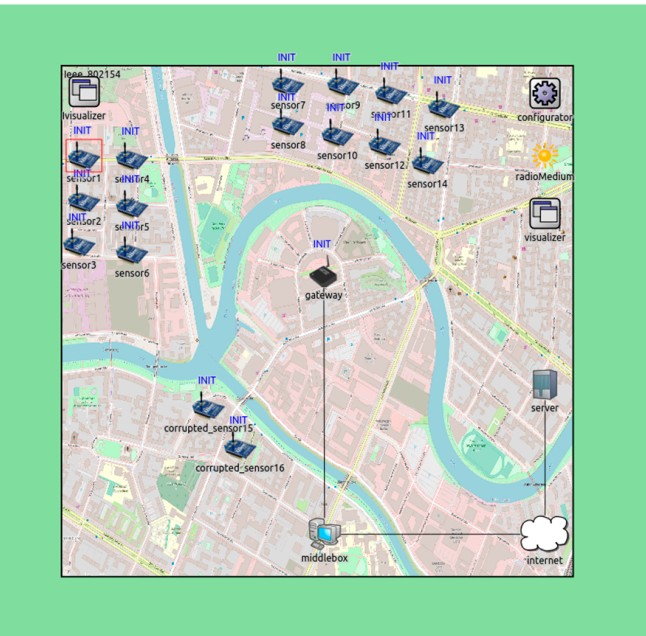

**Figure 6.** General structure of the use case.

The sensors can be of different types (e.g., LoRaWAN or IEEE 802.15.4) which are provided by the OMNET++ simulation models and extensions to the corresponding INET framework [31].

Furthermore, an IoT gateway can be seen in the middle of the network architecture. This is the component to which all the sensors communicate their measurements. The gateway aggregates the traffic from the sensors and places it in IP packets that can be communicated over the Internet to the backend, which would be an IoT platform or an open urban platform as described above. Right behind the gateway, a middlebox can be placed, which is meant to provide the means for protecting the overall IoT and OUP infrastructure and detecting attacks originating from the sensors and gateway towards the backend or vice-versa.

### 5.2. Attack Setup

It is important to remark that Figure 6 contains two sensors which are presumed to be corrupted and to generate malicious IEEE 802.15.4 traffic at a rate of one frame every 10 s into the IoT infrastructure. A general example of the visualization of the IEEE 802.15.4 IoT traffic is shown in Figure 7, where one can observe the measured values being sent in single packets to the IoT gateway and beyond. In this case, the corrupted sensors would be generating meaningless "measurements" on a rate much higher (This means that the "normal" sensors are sending out IEEE 802.15.4 frames on a standard rate, whilst the corrupted sensors are generating IEEE 802.15.4 frames on an extremely high rate within the OMNET++ simulation. This leads to the IoT gateway losing a lot of energy and eventually failing to perform its tasks.)—one IEE 802.15.4 frame every 10 s—than the one of the sane sensors, i.e., one IEEE 802.15.4 frame every 5 min. This could lead to a DoS attack, for instance, towards the gateway, in order to drain its energy and deny the other sensors from the possibility to convey their measurements to the IoT platform or open urban platform in the backend.

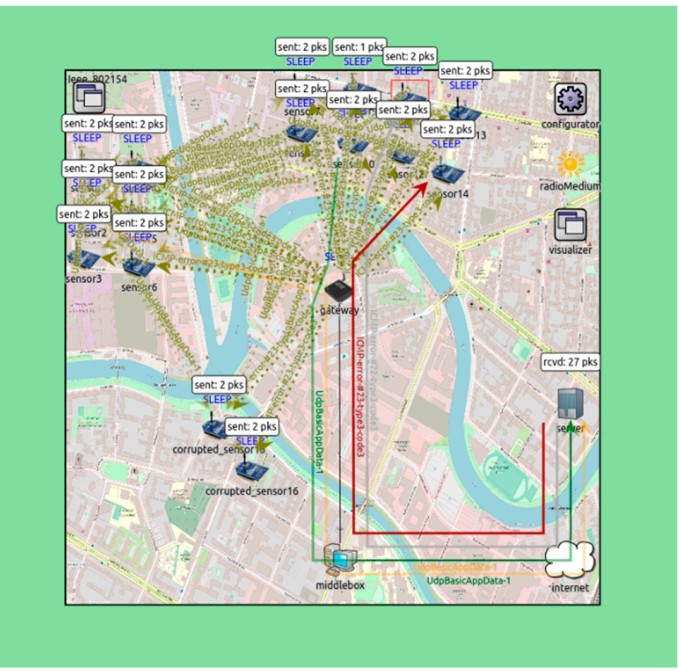

**Figure 7.** IoT traffic visualization.

Figure 8 shows an example from the established simulation, in which the corrupted sensors are successful in making the gateway work so intensively that its battery—provided that the gateway is out in the open without an available power line—quickly drops and degrades the overall performance of the network. Such a situation could have serious consequences for the operator of a critical infrastructure or a mobility service within a smart city. In the case a DoS attack towards the gateway is successful and manages to drain all its energy (thereby effectively shutting it down), then the dependent service or critical infrastructure will be missing important context data and is likely to perform under the level of agreed SLAs (service level agreements), leading, potentially, to the loss of revenue or even compromising the safety of citizens.

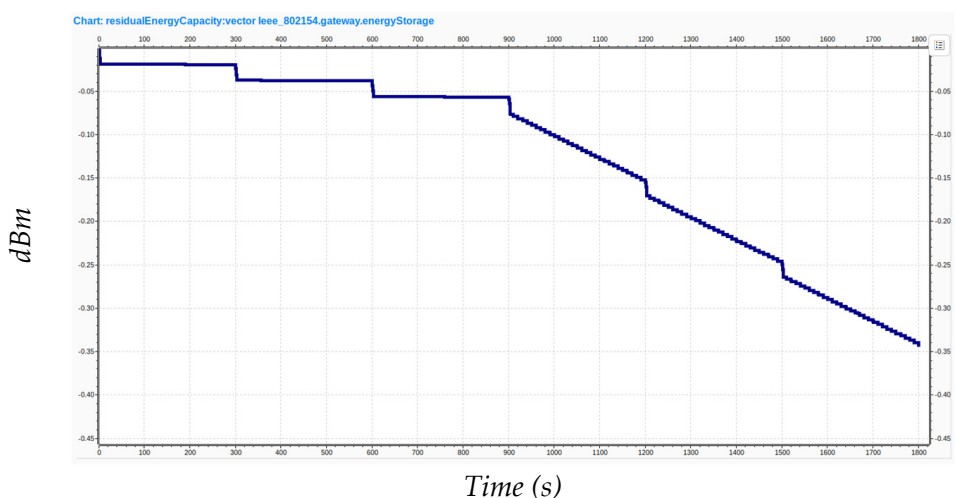

*Time (s)*

**Figure 8.** Residual energy of the gateway—an example from the OMNET++ experiments.

### 5.3. Intrusion Detection

In order to experiment and test the methods for preventing such DoS attacks, traffic was recorded from the simulation and various mechanisms were investigated regarding how to recognize malicious traffic and the corresponding rogue sensors. Such mechanisms

can be either placed directly in the middlebox (see Figures 6 and 7) or can be executed in the according NOC (network operations center) or SOC (security operations center) once the traffic has been monitored and recorded in a real network.

The methods that can be applied for analyzing these data include different autoencoder and deep neural network architectures as well as the concept of a random forest that was mentioned before. In this line of thought, Figure 9 shows the current demonstrator that is based on the DoS recognition utilizing a random forest implemented in Python scikit-learn [32].

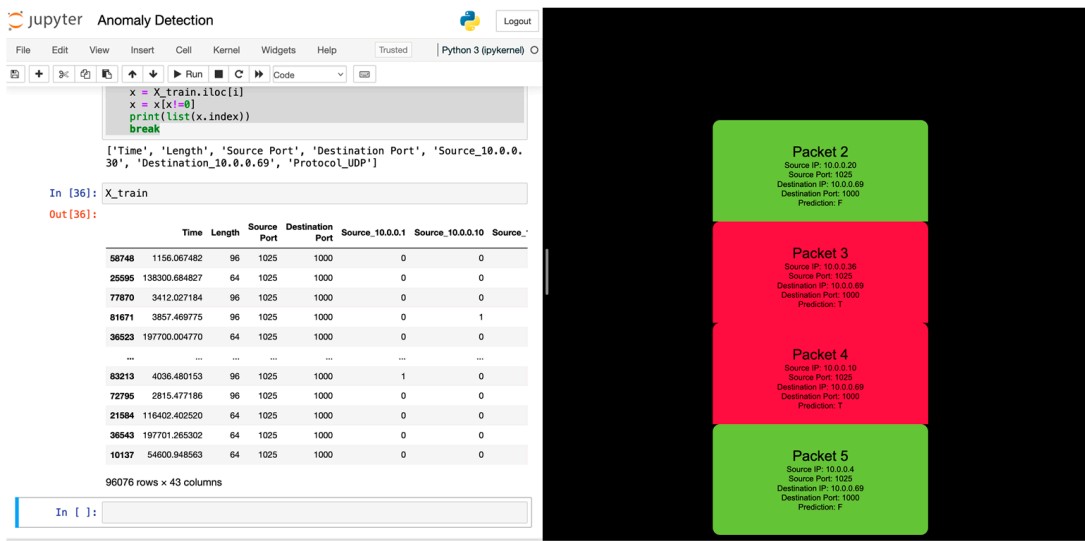

**Figure 9.** Visualization of the demonstrator based on random forest.

The random forest classifier trained in the scope of this work is configured with the main hyperparameters listed in Table 2. The first parameter "*n_estimators*" defines the number of decision trees in the forest. By increasing the number of trees, we might be able to improve the model accuracy, but this could also lead to longer training time and increased risk of overfitting. Therefore, we used the default value of 100 decision trees which provides a good balance between accuracy and computational efficiency. In terms of the split/decision criterion, the random forest trained in this work utilizes the so-called *gini impurity*. *Gini impurity* [33]—a mathematical coefficient determining the degree to which a set consists of a particular type of data/labels—is a good split criterion for a decision tree because it is a fast and efficient way to measure the quality of a split and is effective at identifying the most frequent class in a dataset. The next parameter used for configuring the RF is *max_depth*. This parameter controls the maximum depth of each decision tree in the forest. For the purposes of our use case, we set the parameter to "*None*", which sets no depth limit. The reason is that while maximum depth can help with preventing overfitting, it can also decrease accuracy if the depth is set too low. The final specified parameter is *min_samples_split*, which controls the minimum number of samples required to split an internal node. We set the parameter to be equal to two. This means that a node must have at least two samples to be split. Increasing this value can help to prevent overfitting, but can also lead to a model that is too simple and has lower accuracy.

**Table 2.** Summary of the used Random Forest Parameter.

| Parameter | Value |
|---|---|
| n_estimators | 100 |
| criterion | "gini" |
| max_depth | None |
| min_samples_split | 2 |

*5.4. Overall Demonstrator*

The belonging classification code is provided and showcased as a Jupyter notebook (Figure 9) and is complemented by a self-developed visualization on the right in Figure 9. The visualization shows how the data recorded behind the gateway (e.g., on the middlebox) are sequentially processed and analyzed. Thereby, the sending patterns of the involved sensors are classified by the previously trained random forest, leading to the recognition of erroneous sensors, the packets of which are marked red on the right side in Figure 9.

Indeed, we can observe how the proposed methodology—especially including the network simulation—can be used to model smart city infrastructure (especially IoT network) in question and to develop specific algorithms that can be finally applied in real urban environments. Thereby, the methodological structures and guidelines provided in this paper establish a framework in which urban IoT networks can be systematically protected and improved in terms of cybersecurity capabilities.

## 6. Conclusions

The consistent increase in the number of IoT devices and their major involvement in critical day-to-day tasks (e.g., healthcare, autonomous vehicles, etc.) raises concerns about the security guarantees that these devices can provide. These concerns are based on the observation that, because of their hardware limitations and heterogenous software, IoT devices are vulnerable to both known and unknown attacks.

In this work we presented an overview of the IoT domain and explored some of the attacks and corresponding defenses in the scope of urban IoT networks. This analysis motivated the need for deriving KIVEP—an integrated process and toolchain for setting up and continuously improving cybersecurity solutions in the context of urban IoT networks. Hence, we described a structured KIVEP process that was exemplified on the development of an ML-based anomaly detection mechanism, which was demonstrated and validated in an OMNET++ simulation based on our collaboration with industrial partners and municipalities/utilities in a German research project. The envisioned outcome of the execution of the process laid out in this work assumes that the defense strategies—proposed based on the conducted research efforts—are not only theoretically tested but also practically applied in a production IoT network.

**Author Contributions:** D.R. conducted the overall review of IoT threats and mitigations and created the demonstrator incorporating different ML defense techniques. P.L. worked out the methodology. L.B. contributed to the sections on open urban platforms and facilitated the data acquisition from the industrial partner. S.B. contributed to the sections on the [ui!] platform, obtained the network architecture for the experimentations within the industrial partner premises and reviewed the document. P.D. created the simulation environment and delivered the corresponding figures. N.T. co-defined the methodology, prepared the evaluation/demonstration section, and finalized the document. M.B. supported the ML algorithm development and reviewed the final version of the document. All authors have read and agreed to the published version of the manuscript.

**Funding:** This work was funded by the German Federal Ministry of Education and Research under the funding number V7KMU20/12. The authors are responsible for the content of this publication.

**Data Availability Statement:** Not applicable.

**Conflicts of Interest:** The authors declare no conflict of interest.

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
