# Peer review of "Towards an Integrated Methodology and Toolchain for Machine Learning-Based Intrusion Detection in Urban IoT Networks and Platforms"

_futureinternet, doi:10.3390/fi15030098_

Round 1

Reviewer 1 Report

Review:

In this manuscript, the authors presented a brief overview of existing attacks and defense strategies against urban IoT networks.

Weakness: The manuscript requires revision based on identified issues present in the present version.

1.     According to the title of the manuscript, the keywords is not matching in the Abstract.

2.     It is a survey paper or proposal paper. Confusing. If it is a survey paper, then the survey paper should have comparison tables and various taxonomies. Based on the abstract, it is like a survey……Please revise and justify.

3.   In section 1, motivation is weak….and you did not mention contribution and organization in detail. Mention it.

4.  Line No. 77, What is [ui!]????

5. What is your novelty? It is a major problem in the present version manuscript.

6. Section 2 is confusing. In section 3, you should add a table or taxonomy for existing attacks, then what is the solution to address attacks. Draw the fig. or table. Categorize the attacks also based on existing research and environment.

7. What is the name of the methodology in section 4? Section 4 should be revised based on the present work.

8.  Literature and survey part is also not properly based on the title. Revise it.

9.  What is the future scope of the present work? It should be added in the conclusion part.

10. Carefully check grammatical and typos mistakes. Some syntax is wrong.

11. Cross-reference all citations and ensure that they match accordingly. The reference paper format should be uniform. I have identified some of the references with missing details like page numbers, volume numbers, issue numbers, etc. Recent reference must be added, the following is recommended:

·   Ransomware-based Cyber Attacks: A Comprehensive Survey

·   A survey of cloud-based network intrusion detection analysis for WSNs 

·     A comprehensive analyses of intrusion detection system for IoT environment

I will not recommend further steps based on the organization of the manuscript. But I encourage the authors that the revised version manuscript can resubmit on the portal.

Author Response

>>The manuscript requires revision based on identified issues present in the present version.

>>1.     According to the title of the manuscript, the keywords is not matching in the Abstract.

We added the following keywords: "Methodology, Intrusion Detection, Toolchain"

>>2.     It is a survey paper or proposal paper. Confusing. If it is a survey paper, then the survey paper should have 
>> comparison tables and various taxonomies. Based on the abstract, it is like a survey……Please revise and justify.

This is actually a proposal paper. For this reason, we reworked the abstract heaviliy to avoid confusion. The current
abstract is as follows:

"The constant increase in volume and wide variety of available Internet of Things (IoT) devices leads 
to highly diverse software and hardware stacks, which opens new avenues for exploiting previously unknown vulnerabilities. 
The ensuing risks are amplified by the inherent IoT re-source constraints both in terms of performance and energy expenditure.  
At the same time, IoT devices often times generate or collect sensitive, real-time data used in critical application 
scenarios (e.g. health monitoring, transportation, smart energy, etc.). All these factors combined make IoT networks a 
primary target and potential victims for malicious actors. In this paper, we present a brief overview of existing attacks 
and defense strategies and use this as motivation for proposing an integrated methodology for developing protection mechanisms 
for Smart City IoT networks. The goal of this work is to lay out a theoretical plan and a corresponding pipeline of 
steps - i.e. development and implementation process - for the design and application of cyber-security solutions for urban 
IoT networks. The end goal of following the proposed process is the deployment and continuous improvement of appropriate 
IoT security measures in real-world urban IoT infrastructures. The application of the methodology is exemplified on 
an OMNET++ simulated scenario, which is developed in collaboration with industrial partners."

We also took care of specifying the contribution and the presented validation in a more clear way, thereby 
laying out our approach to the definition and demonstration of the proposed process.

>> 3.   In section 1, motivation is weak….and you did not mention contribution and organization in detail. Mention it.
Two additional sections were added: "1.4. Contribution" and "1.5. Structure of the Presentation".
These section clearly state the contribution of our work and lay out the structure of the paper.

>> 4.  Line No. 77, What is [ui!]????
We added the following explanation in a footnote:
"[ui!] is the abbreviation for Urban Instutute GmbH – this is the industrial partner, with whom the case studies are being investigated and researched."

>> 5. What is your novelty? It is a major problem in the present version manuscript.
A paragraph regardin the novelty was added in section "1.4 Contribution".

"The novelty of our work is provided by the devised methodology for planning and 
continuously improving the setup of cybersecurity solutions for urban IoT networks. 
Our approach is developed in close collaboration with industrial partners and municipality, 
which intrinsically increases its relevance for real-world use cases. Furthermore, the approach 
is validated in an OMNET++ simulation, in order to demonstrate its feasibility and to capture 
feedback from relevant stakeholders (industry and utility companies) in the scope of the KIVEP project [25]."

>> 6. Section 2 is confusing. In section 3, you should add a table or taxonomy for existing attacks, 
>>then what is the solution to address attacks. Draw the fig. or table. Categorize the attacks also based on existing research and environment.

We added a short paragraph at the beginning of section 2, in order to better clarify its role:

"This section focusses on shortly describing the general risks, at which urban IoT networks are exposed. 
Furthermore, it prepares the reader for possible activations of theses risks in terms of potential attacks 
on urban IoT networks. These potential attacks are listed and described together with possible countermeasures in the following section 3."

We added a table categorizing the attacks, defenses and the belonging layers.
Furthermore, the text at the beginning of section 3 was extended to describe the new table.

>> 7. What is the name of the methodology in section 4? Section 4 should be revised based on the present work.

The methodology was named "KIVEP". A corresponding sentense and explanation is provided at the beginning of section 4.

We added a paragraph at the beginning of section 4, which gives the appropriate argumentation as to way the KIVEP-methodology
was devised as is, based on our industrial collaboration:

"The KIVEP methodology is devised as close as possible to the needs and requirements of real-world deployments. 
This statement is based on the fact that we work in close collaboration with key players in the Smart City IoT 
market in Germany as well as with the utility companies and municipalities, which are their direct clients 
in the context of differ-ent IoT deployments. As previously mentioned, this approach intrinsically increases 
the relevance of the below-presented KIVEP methodology for real-world use cases."

>> 8.  Literature and survey part is also not properly based on the title. Revise it.
We added additional literature.

For example, the list of reference was extended by the publications proposed by the reviwer.

>> 9.  What is the future scope of the present work? It should be added in the conclusion part.

We extended the conclusion by further statements showing the scope of the current work:

"In this work we presented an overview of the IoT domain and explored some of the attacks 
and corresponding defenses in the scope of urban IoT networks. This analysis motivated the need for deriving KIVEP 
– an integrated process and toolchain for setting up and continuously improving cybersecurity solutions 
in the context of urban IoT networks. Hence, we described a structured process leading which was 
exemplified on the development of an ML based anomaly detection mechanism, which was demonstrated 
and validated in an OMNET++ simulation based on our collaboration with industrial partners 
and municipalities/utilities in a German research project"

>> 10. Carefully check grammatical and typos mistakes. Some syntax is wrong.

We did a final check on the grammar.

>>11. Cross-reference all citations and ensure that they match accordingly. The reference paper format should be uniform. I have identified some of the references with missing details like page numbers, volume numbers, issue numbers, etc. Recent reference must be added, the following is recommended:
>>·   Ransomware-based Cyber Attacks: A Comprehensive Survey
>>·   A survey of cloud-based network intrusion detection analysis for WSNs 
>>·     A comprehensive analyses of intrusion detection system for IoT environment

We added these publications to the list of references and reffered them in the text, especially in the parts on Machine Learning and Blockchain. 

Reviewer 2 Report

The paper is very interesting. However, presentation should be improved and some questions answered more clearly;

  1. Is the research a real experiment or a simulated one? If it's a real experiment, please add some blueprints of the implementations; pictures, location, etc.
  2. Please provide some tables of the data which devices generated.
  3. Include the traffic data visualization or table which was recorded for investigation.
  4. The graph in figure 8 should be more descriptive.
  5. Provide more information about Autoencoder, Deep neural network, and random forest that are applied; input and output of the algorithm feed.
  6. Include a graphical analysis of the DoS attack detection; graph, confusion matrix, etc.
  7. Remove the picture of the Jupyter notebook code.

Author Response

>>The paper is very interesting. However, presentation should be improved and some questions answered more clearly;

>> Is the research a real experiment or a simulated one? If it's a real experiment, 
>> please add some blueprints of the implementations; pictures, location, etc.

The experiment is a simulated one. However, the focus of the paper is not the experiment, but
the methodology, which is proposed. The experiment is actually an OMNET++ simulation that 
we use as a demonstrator in our lab. The demonstrator is used to show the feasability of
the methodology.

>> Please provide some tables of the data which devices generated.
>> Include the traffic data visualization or table which was recorded for investigation.

The simulated traffic was based on IEEE 802.15.4 frames sent out by the sensors at different rates, namely
at a "normal" rate and at a "corrupted" high rate. In order to better describe this, we added at a couple places
in section 5 that IEEE 802.15.4 frames are being generated. Furthermore, the text was updated to explicitly mention
the rates at which the "normal" and the corrupted packets were sending out the framews.

In addition, we added a footnote for better explanation, saying that:

"This means that the “normal“ sensors are sending out IEEE 802.15.4 frames on a standard rates, whilst the 
corrupted sensors are generating IEEE 802.15.4 frames on an extremely high rate within the OMNET++ simulation. 
This leads to the IoT gateway loosing a lot of energy and eventually failing to perform its tasks."

>> The graph in figure 8 should be more descriptive.

We added the measurement units on the side, in order to clarify the decrease in the residiual energy.
The text also contains the belonging packet generation rates for normal and corrupted sensors.

>> Provide more information about Autoencoder, Deep neural network, and random forest that are applied; input and output of the algorithm feed.

Since the actual demonstrator works based on a random forest classifier (the other two algorithms was just experimented with), we 
added a table (Table 2) with the parameters of the classifier as well as a paragraph describing the meaning of these parameters.

>> Include a graphical analysis of the DoS attack detection; graph, confusion matrix, etc.

Since the evaluation section focusses on demonstrating the process chain and not not experiments relating to the algorithm,
we would like to remain with the current figures which are showing and describing our concrete demonstrator.

>> Remove the picture of the Jupyter notebook c

We would like to keep it for the following reasons: 1) It shows the demonstrator that we have in our lab, 2) the other reviewers were okey with it.

Reviewer 3 Report

The paper outlines a theoretical plan for the deployment of urban IoT infrastructure while anticipating security attacks within the proposed IoT networks. The networks and evaluation in this paper were done using simulation tool the OMNET++. The DoS attacks were simulated and detected using the random forest classification method. Following are comments for the paper:

1. Abstract should also include results and analysis of the overal experiment and state the contribution of the paper clearly.

2. Although the goal of the paper has been well-stated, most part of the paper was dedicated for shallow literature review, i.e., Section 1 to Section 3. In fact, some of them might not be necessary, such as literature review about blockchain that is never used in the study.

3. The claim that the work has tested a DoS attacks migh be misleading. The author didn't show DoS or DDoS type of attacks clearly in the experiment, how they were simulated, how the traffic were generated, and how the effect of the attacks to the system. Although Figure 8 shows the residual energy of the gateway, it doesn't show clearly that this is the effect of DoS or DDoS. Residual energy might be degraded due to many other reasons. The author might simulate the energy degradation by, for example, varying the number and size of the arriving DoS packets in a period of time.

4. The claim of implementing intrussion detection method using the ML and DL methods to detect attacks to the network is also trivial. For example, the random forest method shown in the paper has been widely-used in the previous works for traffic classifications.

5. I guess the author should define the contribution of the paper clearly and start to reconstruct the paper accordingly.

Author Response

>>The paper outlines a theoretical plan for the deployment of urban IoT infrastructure while anticipating security attacks within 
> the proposed IoT networks. 
>>The networks and evaluation in this paper were done using simulation tool the OMNET++. 
>>The DoS attacks were simulated and detected using the random forest classification method. Following are comments for the paper:

>>1. Abstract should also include results and analysis of the overal experiment and state the contribution of the paper clearly.

We updated the abstract in order to sharpen the goal of the paper and to show the role of the experiment as a validation
and demonstration of the methodology.

Abstract update at the end:
"The application of the methodology is exemplified on an OMNET++ simulated scenario, which is developed in collaboration with industrial partners."

>>2. Although the goal of the paper has been well-stated, most part of the paper was dedicated for shallow literature review, i.e., Section 1 to Section 3. In fact, some of them might not be necessary, such as literature review about blockchain that is never used in the study.

We added statements in order to show that the literature overview is used as a motivation for the developed methodology.
In this manner, we show why we need the overview of different attacks and countermeasures - namely to argument that we 
need a continuous integrated approach for setting up cybersecurity solutions and improving their operation over time.

The following updates were made in this regard:
* Abstract: "In this paper, we present a brief overview of existing attacks and defense strategies and 
use this as motivation for proposing an integrated methodology for developing protection mechanisms for Smart City IoT networks." 

* Added section 1.4 contribution. The second bullet point there shows the role of the literature overview.

* Added section "1.5 Structure of the presentation" --> especially the following statement:

"Section 3 continues with a more detailed discussion of the potential security issues and summarizes available 
approaches to increasing the level of IoT cybersecurity in urban ICT. The general issue that stands out is that 
all these single approaches do not represent an overall picture of how cybersecurity should be addressed in the 
Smart City context for IoT networks."

* Added Table 1 with the classification of Attack Types and Countermeasures, which also is discussed 
in the text as a motivation for the methodology

>>3. The claim that the work has tested a DoS attacks migh be misleading. The author didn't show DoS or DDoS type of 
>>attacks clearly in the experiment, how they were simulated, how the traffic were generated, and how the effect of the attacks to the system. 

We extended the text by statments saying that the devices generate IEEE 802.15.4 frames on particular rates - one packet every 5 minutes
for "normal" sensors and one packet every 10 seconds for corrupted/erroneous sensors.

>>Although Figure 8 shows the residual energy of the gateway, it doesn't show clearly that this is the effect of DoS or DDoS. 
>>Residual energy might be degraded due to many other reasons. 
>>The author might simulate the energy degradation by, for example, varying the number and size of the arriving DoS packets in a period of time.

We agree that this is possible, however the figure is a clear observation as a result of the higher packet rates from the erroneous
sensor nodes within the simulation. After we started those, we observed the depicted decline in the residual energy.
These interrelations are also elucidated on in the text.

>>4. The claim of implementing intrussion detection method using the ML and DL methods to detect attacks to the network is also trivial. 
>>For example, the random forest method shown in the paper has been widely-used in the previous works for traffic classifications.

The claim is not that the we introduce novelty by using the random forest for classification, but that we demonstrate the proposed methodology
in the simulated environment based on a real-world use case and ML classificators

This is especially addressed in section 1.4 by the following paragraph:

"The novelty of our work is provided by the devised methodology for planning and continuously improving the 
setup of cybersecurity solutions for urban IoT networks. Our approach is developed in close collaboration 
with industrial partners and a municipality, which intrinsically increases its relevance for real-world use cases. 
Furthermore, the approach is validated in an OMNET++ simulation, in order to demonstrate its feasibility and to 
capture feedback from relevant stakeholders (industry and utility companies) in the scope of the KIVEP project [25]."

>>5. I guess the author should define the contribution of the paper clearly and start to reconstruct the paper accordingly

The paper was correspondingly updated in accordance to the comments of the reviewer and the other evaluators.
Thank you for the useful feedback :-)

Reviewer 4 Report

Based on the abstract, the goal of this paper is to present (i) a summary of common IoT attacks and solutions and (ii) to introduce a pipeline for the implementation of urban IoT networks and platforms. While the presentation of this paper is complete, the information given is still fragmented and unorganized.

The authors should describe the background/motivation, related works, proposed method, evaluation results, and conclusion in a nice information flow from start to finish.

Section 3 supposedly becomes the authors' idea to solve goal (i), but there is no correlation between the given attacks and the given solutions (through ML/blockchain); the authors feed the readers much information without any guidelines of why it is described.

Secondly, Section 3's contents are not used or have a direct relationship with Sections 4 and 5; hence, it is unclear why this discussion is brought up in the first place. The experiments in Section 5 focus on DoS mitigation using random forests. Then why are we given information about many IoT attacks/solutions, ML/blockchains in Section 3?

Sections 1.1. to 1.3, Section 2, and Section 3 can be reorganized into a single Section as a "bridge/background" for the proposed template/pipeline in Sections 4 and 5.

There is no discussion on similar urban IoT network planning.

The conclusion should mention what "result/experience" the authors gain from the proposed method and evaluation.

Other minor comments include the following:

In Figure 1, the "internet vs. cellular" in the "IoT platform & connectivity" does not accurately represent this topic. The cellular is part of the Internet and should not be compared side by side. Furthermore, the CEP (Complex Event Processing) acronym can be put directly in the figure since it is not a common abbreviation; this way, the readers can know what it means without reading the paragraph.

The authors can end the "Introduction" section by mentioning the roadmap of the paper, such as "The rest of this paper is organized as follows. Section 2 describes A. Section 3 discusses B, and so on."

Figure 2 and Figure 3 can be merged.

The title of Section 3, "Discussion," does not accurately represents the contents. The authors should rename this title.

In Section 4, each box in Figure 4 can be explained nicely in one subsection or one paragraph.

What are the x-axis and y-axis in Figure 6?

Author Response

>>Based on the abstract, the goal of this paper is to present (i) a summary of common IoT attacks and solutions and 
>> (ii) to introduce a pipeline for the implementation of urban IoT networks and platforms. 
>> While the presentation of this paper is complete, the information given is still fragmented and unorganized.
>> The authors should describe the background/motivation, related works, proposed method, evaluation results, and conclusion in a 
>> nice information flow from start to finish.

Thank you for your comment.
We reworked the paper at several places in order to address your comment.
This includes updates to the abstract, the addition of section 1.4 and 1.5 in order to show and present
the structure of the presentation as well as updates to the conclusion with the goal to clearly
show the intention of the paper and how it was achieved/validated.

>> Section 3 supposedly becomes the authors' idea to solve goal (i), but there is no correlation between the given 
>> attacks and the given solutions (through ML/blockchain); the authors feed the readers much information without any guidelines of why it is described.

Section 3 was wrongly named: It's new title is "Discussion and Classification of IoT Attacks and Countermeasures"

Section 3 is the discussion and deepening of the problem statement thereby detailing potential attacks and solutions.
We updated the beginning of "Section 2 Problem Statement" in order to clearly show this.

"This section focusses on shortly describing the general risks, at which urban IoT networks are exposed. 
Furthermore, it prepares the reader for possible activations of theses risks in terms of potential attacks on urban IoT networks. 
These potential attacks are listed and described together with possible countermeasures in the following section 3."

In addition, we added "Table 1: Classification of Attack Types and possible Defenses or Architectural Measures in Urban IoT Networks".
This puts the different attacks in realtion to the IoT layer and the possible solution. The goal of our proposed 
methodology is to provide the structure for setting-up such solutions and continuously updating and improving them
in order to increase the overall cybersecurity of the system

>> Secondly, Section 3's contents are not used or have a direct relationship with Sections 4 and 5; hence, 
>> it is unclear why this discussion is brought up in the first place. 

The section 3 contents are the motivation for section 4 and section 5. We changed the name of section 3 and added
statements in the abstract and introduction to clarify this.

The new Table 1 shows the relation between the solutions and the attacks.
Hence, the usage of ML as a method is argued for there.

>> The experiments in Section 5 focus on DoS mitigation using random forests. 
>> Then why are we given information about many IoT attacks/solutions, ML/blockchains in Section 3?

The demonstrator in section 5 is only meant to show one example of how the tool chain can be instantiated.
Several sections were updated or added to make this clear. Concretely, we added section 1.4 that clearly
shows the novelty and contribution of the paper while arguing that we validate the proposed methodology
through the demonstrator in section 5.

>> Sections 1.1. to 1.3, Section 2, and Section 3 can be reorganized into a single Section as a "bridge/background" for 
>>the proposed template/pipeline in Sections 4 and 5.

Since the other reviewers were okey with the current structure, we would prefer to keep it as is.

>>There is no discussion on similar urban IoT network planning.

The way we derived our approach was based on the collaboration with industry and municipalities in the KIVEP project.
Hence, it is based on real-world requirements regarding the security side of IoT infrastructure planning.

The statement was placed in "1.4 Contribution":

"The novelty of our work is provided by the devised methodology for planning and continuously improving the setup of 
cybersecurity solutions for urban IoT networks. Our approach is developed in close collaboration with industrial partners 
and a municipality, which intrinsically increases its relevance for real-world use cases."

Hence, our approach is more requirements driven based then rooted in the scientific world.

>>The conclusion should mention what "result/experience" the authors gain from the proposed method and evaluation.

 We updated the conclusion as follows:

"In this work we presented an overview of the IoT domain and explored some of the attacks and corresponding defenses in the scope of urban IoT networks. 
This analysis mo-tivated the need for deriving KIVEP – an integrated process and toolchain for setting up and continuously improving cybersecurity solutions 
in the context of urban IoT networks. Hence, we described a structured process leading which was exemplified on the development of an ML based anomaly 
detection mechanism, which was demonstrated and vali-dated in an OMNET++ simulation based on our collaboration with industrial partners and 
municipalities/utilities in a German research project. The envisioned outcome of the execution of the process laid out in this work assumes that 
the defense strategies - proposed based on the conducted research efforts - are not only theoretically tested but also practically applied in 
a production IoT network."

>> Other minor comments include the following:

>> In Figure 1, the "internet vs. cellular" in the "IoT platform & connectivity" does not accurately represent this topic. 
>> The cellular is part of the Internet and should not be compared side by side. 

The reviewer is only partially right. The Internet includes everything that runs the IP protocol.
In the cellular part we very often don't have IP running. Good examples are given by old GSM and UMTS
as well as by IoT technologies such as IEEE 802.15.4 and LoRaWAN.

>> Furthermore, the CEP (Complex Event Processing) acronym can be put directly in the figure since 
>> it is not a common abbreviation; this way, the readers can know what it means without reading the paragraph.

The reviewer is only partially right. CEP is a very common term: https://en.wikipedia.org/wiki/Complex_event_processing
It was first used in 2000-2010 in the course of P2P networks and network and systems management.
There is a variety of frameworks for this task with Kafka being the most famous one.

>> The authors can end the "Introduction" section by mentioning the roadmap of the paper, such as 
>> "The rest of this paper is organized as follows. Section 2 describes A. Section 3 discusses B, and so on."

We added section "1.5. Structure of the Presentation".

>> Figure 2 and Figure 3 can be merged.

We would keep them separately. Figure 2 describes the layers of an IoT-platform, while Figure 3 deals with the 
concept of Edge-Fog-Cloud. These are two different things, eventough they might have an overlap in many cases.

>> The title of Section 3, "Discussion," does not accurately represents the contents. The authors should rename this title.

Section 3 was renamed to "3. Discussion and Classification of IoT Attacks and Countermeasures".

>> In Section 4, each box in Figure 4 can be explained nicely in one subsection or one paragraph.

We changed the names of the sections and added some text and a new subsection - we hope that we could fulfill your request.
The captions of the subsections are not exactly the titles of the boxes, but each subsection
explains one of the components in the figure.

>> What are the x-axis and y-axis in Figure 6?

Figure 6 does not contain any relevant x- or y-axis. We updated Figure 8 with the belonging captions for the x- and y-axis.

Round 2

Reviewer 2 Report

The revised paper can be published as it is

Author Response

>> The revised paper can be published as it is

Thank you :-)

Reviewer 3 Report

Authors revised most part of the manuscript and made it in line with the title and the goal stated. Minor comments for the paper are following:

1. It might useful to elaborate a little bit about the KIVEP project.

2. Several figures are not well numbered and there might be some wrong figure citations within paragraphs due to change on the order of the figures and some additional figures in the manuscript. 

Author Response

>>1. It might useful to elaborate a little bit about the KIVEP project.

We added a subsection (section 1.4 in the introduction) briefly describing the KIVEP project.

>>2. Several figures are not well numbered and there might be some wrong

>>citations within paragraphs due to change on the order of

>> the figures and some additional figures in the manuscript. 

We checked the figures numbering and corrected the references in the text.

Reviewer 4 Report

The authors have addressed my previous comments well.

Author Response

>> The authors have addressed my previous comments well.

Thank you :-)